# Counterfactual Regret Minimization for Sequential Equilibrium

## Abstract

Computing the *Nash equilibrium* (NE) in the imperfect-information two-player zero-sum sequential game is an important problem. Finding some refinements of the Nash equilibrium is important because the Nash equilibrium may take suboptimal actions in states that can not be reached in equilibrium. In this work, we improve the framework of the *counterfactual regret minimization* (CFR) algorithm, proving that our algorithm can converge to the refinements of the Nash equilibrium under some assumptions. The *extensive-form perfect equilibrium* (EFPE) and the *sequential equilibrium* (SE) are two refinements of the Nash equilibrium, they improve on this shortcoming of the Nash equilibrium by assuming that players make mistakes. Most current sequential equilibrium and extensive-form perfect equilibrium computing algorithms are not iterative algorithms and need to solve linear programs, which are ineffective on large-scale games. Our method gives a local perturbation in all the states in the game and gives a suitable perturbation descent method. We compare our *Sequential Perturbed Counterfactual Regret Minimization* (SPCFR) algorithm with CFR variants and the approximate EFPE computing algorithm, perturbed CFR. Experimental results show that our method outperforms existing CFR-based methods on popular games, including Kuhn Poker, Leduc Hold'em, and GoofSpiel.

## 1 Introduction

Computing the *Nash equilibrium* (NE) Nash (1951) in the imperfect-information two-player zero-sum sequential game is a great challenge for artificial intelligence. Counterfactual regret minimization (CFR) Zinkevich et al. (2007) and policy space response oracles (PSRO) Lanctot et al. (2017) are two successful frameworks in equilibrium computing. PSRO is based on Empirical Game Theoretic Analysis(EGTA) Wellman (2006) that samples strategies from large extensive-form games to the metagame. Meanwhile, CFR minimizes the regret value in all the nodes in the game tree to compute the NE. Recent research has also produced many superhuman intelligences that can defeat humans in games such as Go Silver et al. (2016) and Texas Hold'em Brown & Sandholm (2018).

The NE is an equilibrium in which no player can increase their payoff simply by changing their strategy, which assumes that other players always rationally follow the NE strategy. This assumption is not reasonable, especially when our agent faces human opponents who often cannot take optimal actions. When facing such irrational opponents, if the game reaches states (information sets) that are unreachable in the NE, the equilibrium strategy typically cannot take optimal actions in these situations.

Researchers have proposed several refinements to address the limitations of NE in imperfect information extensive-form games, particularly its inability to take optimal actions at unreachable information sets. Notably, the *extensive-form perfect equilibrium* (EFPE) Selten (1975) and *sequential equilibrium* (SE) Kreps & Wilson (1982) are two such refinements that enhance the predictive power of game-theoretic models in scenarios involving irrational or suboptimal opponents. The EFPE defines a perturbed game, which ensures all actions have a minimum positive probability. It tolerates players making unexpected moves, known as "trembles". The limit of the perturbed game's equilibrium as this perturbation tends to zero is the EFPE. The SE refines the NE by considering players' beliefs about which node in an information set they are at, especially when that information set is unreachable under equilibrium strategies. The SE gives two conditions for equilibrium: sequen-

tial rationality and consistency. This ensures that players' strategies remain optimal even off the equilibrium path.

Recent research has introduced several approaches to computing the refinement of the NE in two-player zero-sum sequential games. Farina et al. integrates perturbations into the CFR framework. However, this method fixes the perturbation size, which cannot compute the accurate EFPE. The fixed perturbation will also limit the exploitability of the strategy. Bernasconi et al. utilizes the Online Optimistic Mirror Descent (OOMD) Rakhlin & Sridharan (2013) algorithm to compute the EFPE. While this approach provides theoretical convergence guarantees, the OOMD algorithm does not perform well in large-scale games. Specifically, this algorithm performs worse than CFR in the Goofspiel environment in the thesis's experiments.

This paper proposes the Sequential Restricted Counterfactual Regret Minimization algorithm. We extend the concept of the perturbed game, transforming the perturbation from a global setting to a local one at each information set. To overcome the problems caused by fixed perturbations, we give a reasonable way to decrease the perturbation during the computation. This condition is necessary for the algorithm to converge to the SE. We also prove it is a sufficient condition for the algorithm to converge to the SE under certain assumptions. We improve the part of regret matching in the CFR algorithm by reducing the perturbation of the game, which we call the *sequential perturbed regret matching* (SPRM). We also give the upper bound on the average regret of the improved algorithm. We compare our algorithm with CFR variants and the approximate EFPE computing algorithm, perturbed CFR, in three environments, Kuhn Poker, GoofSpiel, and Leduc Hold'em. The evaluation metrics we used are exploitability and maximum information set regrets.

## 2 RELATED WORK

The variants of *Counterfactual Regret Minimization* (CFR) are the most successful family of algorithms for solving imperfect-information games. MCCFR Lanctot et al. (2009) utilizes Monte Carlo sampling so that the algorithm does not need to traverse the entire tree, increasing the speed of the CFR algorithm in large imperfect-information games. CFR+ Bowling et al. (2015) sets the cumulative regret to 0 directly when it is negative, greatly increasing the convergence speed of the CFR. Deep-CFR Brown et al. (2019) uses neural networks to approximate the behavior of CFR in the full game, which has strong performance in large games. DiscountCFR (DCFR) Brown & Sandholm (2019) adds a discount factor to past regret values when computing the historical average, effectively increasing the convergence efficiency of the CFR.

CFR is not the only iterative algorithm capable of solving the Nash equilibrium of large imperfect-information games. *Neural Fictitious Self Play* (NFSP) Heinrich & Silver (2016) combined deep reinforcement learning with *Fictitious Play* Brown (1951) to solve the Nash equilibrium of large imperfect-information games. However, *Fictitious Play*(FP) has weaker theoretical convergence guarantees than CFR, and converges more slowly in practice.

Due to the problems that the NE may take suboptimal actions in unreachable information sets, several studies are aimed at computing the refinement of the NE. Two of the most commonly used refinements are the *Extensive-Form Perfect Equilibrium* (EFPE) Selten (1975) and the *Sequential Equilibrium* (SE) Kreps & Wilson (1982). Below, we first discuss the work on computing the SE.

Miltersen & Sørensen firstly uses the minimax strategies to compute the SE for two-player games and proves that the SE of two-player zero-sum games can be solved in polynomial time. Gilpin & Sandholm gives the method to compute the SE in games where chance nodes are the only source of uncertainty. Turocy provides a numerical algorithm to compute the SE for finite imperfect-information games. However, the implementation suffers from numerical instability, making it unreliable Miltersen & Sørensen (2006b). Panozzo proposed some approaches for the algorithmic verification of the SE. Finally, Graf et al. gives a symbolic solution to the SE of the general form game. However, there is no method for solving SE using an iterative algorithm.

For finding the EFPE, another refinement of the NE, the first works in two-player zero-sum sequential games are presented by Hansen et al.; Farina & Gatti; Farina et al.. However, all these works are based on solving *linear programs* (LP) Dantzig (1963) to compute the EFPE. Because solving LP is not an iterative algorithm, they fail to address complex real-world problems. Farina et al. gives the first iterative algorithm to compute the EFPE, which is based on the CFR framework. This algorithm

computes the equilibrium in a perturbed game with a fixed perturbation size, which means it cannot compute an exact EFPE. More critically, since the perturbation enforces a minimum probability for each action, the strategy will continuously take the suboptimal actions, leading to a lower bound on exploitability. This limitation prevents the strategy from achieving true optimality. Bernasconi et al. gives the first iterative algorithm that can converge to the EFPE. The OOMD algorithm needs to be optimized on the overall game matrix, not perform well in large-scale games.

# 3 PRELIMINARIES

## 3.1 EXTENSIVE-FORM GAMES

The extensive-form game (EFG) is a widely used game model for multi-player sequential games. In EFGs, there are two important concepts: imperfect information and imperfect recall. Imperfect information means the players may lack some information about the game's state. An example is the Leduc Hold'em, where players cannot see each other's cards. Imperfect recall means players may forget some of the information they have observed. An example is that players forget the opponents' previous play in any card game. In this work, we do not consider the game with imperfect recall. The definition of the imperfect information EFG with perfect recall is as follows.

**Tree Structure.** A finite imperfect-information extensive-form game with perfect recall can be described as a tuple $G :=< N, H, A, V, \mathcal{I}, P, u >$. The components of G are defined as follows,

- $N = \{0, 1, ..., n\}$ is the set of players. Especially, player 0 presents nature(chance node), a nonstrategic agent responsible for all the random events in the game. While all other players are strategic agents.

- $H$ is the finite game tree. Every node $h \in H$ is present by a set $h = \{h_p, a\}$. $h_p$ is the parent node of $h$. $a$ is the edge from the parent node $h_p$ to node $h$, which present the action taken in the node $h_p$. Especially, we define the root node as the empty set $h_0 = \varnothing$.

- $A : H \to 2^{\mathcal{A}}$ gives the legal actions of every node in the game tree $H$, where $\mathcal{A}$ is the set of all possible actions in the game tree. If $A(h) = \varnothing$, we call $h$ a terminal node. $T$ is the set of all terminal nodes $T = \{h | A(h) = \varnothing\}$. Other nodes are decision nodes, represented by $D = \{h | A(h) \neq \varnothing\}$.

- $V : D \to N$ represents which player should take action in the decision node.

- $\mathcal{I} = (\mathcal{I}_1, \mathcal{I}_2, ..., \mathcal{I}_n)$ is the information sets for all player. $\mathcal{I}_j$ is a partition of $V^{-1}(j)$, which satisfied that $\forall I_1, I_2 \in \mathcal{I}_j, I_1 \neq I_2, s.t.\ I_1 \cap I_2 = \varnothing$ and $\bigcup_{I \in \mathcal{I}_j} I = V^{-1}(j)$. For all node $h \in H$ that $V(h) \neq 0$, it belongs and only belongs to an information set $I \in \mathcal{I}_{V(\langle\rangle)}$.

- A node $h$ where $V(h) = 0$ is a chance node. $C = \{h | V(h) = 0\}$ is the set of chance nodes. $P(\cdot | h) : C \to (\mathcal{A} \to \mathbb{R}^+)$ is the action probability distribution in the chance node.

- $u : T \to \mathbb{R}^n$ is the utility function that gives the utility of all normal players in the terminal node.

**Strategies**. In the above definitions, none of the nodes specify the probability distribution of their actions, except for the chance nodes for which we have defined the probability distribution of their legal actions. We call the behavior of the normal player $j \in N \backslash \{0\}$ in the game a strategy.

The pure strategy of player $j$ is $\pi_j(I) : \mathcal{I}_j \to \mathcal{A}$, which specifies that the player $j$ performs a defined action at each information set. The mixed strategy of player $j$ is $\beta_j(\cdot | I) : \mathcal{I}_j \to (\mathcal{A} \to \mathbb{R}^+)$, which gives the probability distribution of legal actions in every information set of player $j$.

**Beliefs**. For each information set, the player needs a probability distribution that determines which node in the information set they are in. We call this probability distribution the *belief*.

The belief is a conditional probability distribution in the information set under the strategy $\beta$,

$$\mu_\beta(h|I) = \frac{P_\beta(h)}{P_\beta(I)}, \tag{1}$$

where $P_\beta(h)$ and $P_\beta(I)$ is the reach probability of node $h$ and information set $I$. $\beta = \{\beta_1, \beta_2, ..., \beta_n\}$ is the strategies of all normal players. If all the players follow strategy $\beta$, the

product of each edge from the root node to node $h$ is the reach probability of node $h$. The reach probability of information set $I$ is the summation of each node $h$ in the information set $I$.

$$P_\beta(h) = P_\beta(h_p)\beta_{V(h)}(a|I_h), \tag{2}$$

$$P_\beta(I) = \sum_{h \in I} P_\beta(h), \tag{3}$$

where $\{h_p, a\} = h$.

**Utilities**. The utility of terminal node $h \in T$, $u(\beta|h) = u(h)$ has already been defined in the game $G$. The utility of other nodes is the expected utility the player will get in these nodes, if all the players follow the strategy $\beta$. The utility of the information set is also the expected utility the player will get according to the belief defined in Equation 1. The definition of utility is

$$u(\beta|h) = \sum_{a \in A(h)} \beta_{V(h)}(a|I_h)u(\beta|\{h, a\}), \tag{4}$$

$$u(\beta|I) = \sum_{h \in I} \mu_\beta(h|I)u(\beta|h), \tag{5}$$

where $I_h$ is the information set where $h$ is located. $\beta_0$ denotes the strategy of nature $P$ for convenience.

## 3.2 Equilibria of the Game

Equilibrium is the computational goal of the game. For a given game, our objective is typically to find an equilibrium. The *Nash equilibrium* (NE) is the most commonly used solution concept.

The NE refers to a situation where neither player can increase their utilities by changing their strategy, so no one is willing to change it. To define the NE, we specify $\beta_{-i}$ as the strategies of all other players except player $i$. $u_i(\beta)$ is the utility of player $i$ when all players use strategies $\beta$. Then the Nash equilibrium can be defined as follows.

**Definition 1. (Nash Equilibrium)** *A strategy profile $\beta$ is a NE if $u_i(\beta'_i, \beta_{-i}) \leq u_i(\beta_i, \beta_{-i})$ for all players $i$ and all strategies $\beta'_i$ of player $i$.*

Although the subsequent sections of this paper do not focus on the *extensive-form perfect equilibrium* (EFPE), some of the algorithmic ideas we employ are related to the EFPE. Therefore, we provide a brief introduction to the EFPE.

A $\varepsilon$-perturbed game means that for all the normal players, the probability of each of their actions must be greater than $\varepsilon$. We call the NE on the $\varepsilon$-perturbed game an $\varepsilon$-EFPE.

**Definition 2. (Extensive-Form Perfect Equilibrium)** *A strategy profile $\beta \in \Pi^\varepsilon$ is an $\varepsilon$-EFPE if $u_i(\beta'_i, \beta_{-i}) \leq u_i(\beta_i, \beta_{-i})$ for all players $i$ and all strategies $\beta'_i \in \Pi^\varepsilon_i$ of player $i$. $\Pi^\varepsilon_i$ and $\Pi^\varepsilon$ can be defined as follows, $\Pi^\varepsilon_j = \{\beta_j | \forall I \in \mathcal{I}_|, \forall a \in A(I), \beta_j(a|I) \geq \varepsilon\}$, $\Pi^\varepsilon = \{\Pi^\varepsilon_j | j \in N \setminus \{0\}\}$.*

*The EFPE is the limitation of the $\varepsilon$-EFPE when $\varepsilon \to 0^+$.*

The sequential equilibrium (SE) is a refinement of the NE tailored for extensive-form games. In SE, a central concept is called the "assessment", the pair of strategy and the belief $(\beta, \mu_\beta)$. The definition of belief is given in Equation 1. It is easy to see that this definition of belief has a problem with unreachable information sets. In order to solve this problem, an assessment $(\beta, \mu_\beta)$ must satisfy two conditions: sequential rationality and consistency. The full definition is as follows,

**Definition 3. (Sequential Equilibrium)** *A strategy profile $\beta$ is a SE if the assessment $(\beta, \mu_\beta)$ satisfied,*

- **sequential rational**. *$u_{V(I)}(\beta'_{V(I)}, \beta_{-V(I)}, \mu_\beta|I) \leq u_{V(I)}(\beta_{V(I)}, \beta_{-V(I)}, \mu_\beta|I)$ for all information sets $I \in \mathcal{I}$ and all strategies $\beta'_{V(I)}$ of player $V(I)$.*

- **consistent**. *There exists a series of assessments $(\beta^n, \mu_\beta^n)$, such that $\lim_{n \to \infty}(\beta^n, \mu_\beta^n) = (\beta, \mu_\beta)$, and for all $h \in H, I \in \mathcal{I}$,*

$$P_{\beta^n}(h) > 0$$

$$\mu_{\beta^n}^n(h|I) = \frac{P_{\beta^n}(h)}{P_{\beta^n}(I)} \tag{6}$$

From the definitions, we can observe that the EFPE is included in the SE. This is because if we choose any sequence of the $\varepsilon$-EFPE where $\varepsilon \to 0$, these sequences satisfy the definition of consistency of the SE.

## 3.3 Counterfactual Regret Minimization(CFR)

CFR is a self-play algorithm using regret minimization. There are two core concepts in the CFR. One is the *counterfactual regret*, and the other is the *regret matching*. There is a well-known theory of the relationship between regret values and Nash equilibrium. If both players' average regrets are less than $\varepsilon$, the average of their historical strategies is an $\varepsilon$-Nash equilibrium Waugh (2009). The $\varepsilon$-Nash equilibrium means players have an $\varepsilon$ tolerance for strategy, which means if any player deviates from the equilibrium, he will not increase the utility more than $\varepsilon$.

*counterfactual regret* is based on *counterfactual probability*. If player $j$ uses a strategy consistent with the historical actions, and other players follow the strategy $\beta_{-j}$, the reaching probability from the root node to this node is the *counterfactual probability* $P^{\beta}_{-j}(h)$.

$$P^{\beta}_{-j}(h) = \begin{cases} 1, & h = \varnothing \\ P^{\beta}_{-j}(h_p), & V(h_p) = j \\ P^{\beta}_{-j}(h_p)\beta_{V(h_p)}(a), & V(h_p) \neq j \end{cases} . \tag{7}$$

$P^{\beta}_{-j}(I)$ is the counterfactual probability of the information set, which is the summation of all the nodes $h$ in the information set $I$,

$$P^{\beta}_{-j}(I) = \sum_{h \in I} P^{\beta}_{-j}(h). \tag{8}$$

We call the multiple of counterfactual probability and the utility of the node the *counterfactual value* $v(\beta|h) = P^{\beta}_{-V(h)}(h)u(\beta|h)$. The concept of *counterfactual regret* can be explained as follows. If the player takes action $a$ instead of following policy $\beta$, the extra counterfactual value the player can get in this information set is the counterfactual regret.

$$r_j(a|I) = P^{\beta}_{-j}(I)(u_j(\beta_{I \to a}|I) - u_j(\beta|I)), \tag{9}$$

where $\beta_{I \to a}$ means in information set $I$, player take action $a$ but follow strategy $\beta$ in all other information set.

*Regret matching* is the method of using historical regrets to generate strategies. We use $R^T_j(a|I) = \sum_{t=1}^{T} r^t_j(a|I)$ present the summation of historical regrets of player $j$ in information set $I$. A new strategy in time step $T + 1$ generated by the regret matching is,

$$\beta^{T+1}_j(a|I) = \begin{cases} \dfrac{[R^T_j(a|I)]_+}{\sum_{a \in A(I)}[R^T_j(a|I)]_+}, & \exists[R^T_j(a|I)]_+ > 0 \\ \dfrac{1}{|A(I)|}, & otherwise \end{cases} , \tag{10}$$

where $[x]_+$ means $\max\{x, 0\}$.

## 4 Method

Perturbed CFR uses the CFR algorithm directly on the perturbed game to compute the approximate EFPE, and we will follow this idea here. Building upon this, we reduce the perturbation at a reasonable rate to refine the NE. Since the perturbed game in our algorithm uses local perturbations (which we will later explain are necessary), proving the algorithm converges to the EFPE is challenging. Therefore, we will prove that the algorithm can converge to the SE under certain assumptions and conditions we have given are necessary for the convergence. Next, we will introduce the method's two parts, the Perturbed Regret Matching and the Sequential Perturb Decreasing.

## 4.1 PERTURBED REGRET MATCHING

This section follows the approach presented in perturbed CFR, but we describe it in our style and include a proof of convergence under variable perturbations. Since the perturbation changes with the information set and the iterations, we use $\delta_T(I)$ to present the perturbation in information set $I$ at iteration $T$.

$$\sigma_j^T(I, a) = (1 - \delta_T(I))\beta_j^T(I, a) + \frac{\delta_T(I)}{|A(I)|}, \tag{11}$$

where $\sigma_i^T(I, a)$ is the strategy follow the regret matching. It is a mixed strategy of regret matching strategy and uniform random strategy, and every action will be chosen with probability at least $\frac{\delta_T}{|A(I)|}$.

We proved that if $\lim_{T\to\infty} \delta_T(I) = 0$ for all information sets $I$, the historical average of strategy $\sigma$ still converges to the Nash equilibrium. The upper limit of average regret follows Theorem 1.

**Theorem 1.** *If any information set $I$ satisfied $\delta_T(I) \sim O(T^\alpha), \alpha < 0$, the upper limit of the regret in this information set is $R_{\sigma,T}(I) \sim O(T^{\max\{0.5, \alpha+1\}})$.*

A full proof of Theorem 1 is provided in Appendix A. From this theory, we know that the asymptotic convergence rate of the CFR algorithm will not be limited if the descent rate of $\delta_T(I)$ is not slower than $T^{-0.5}$.

---

**Algorithm 1** Traversal with perturbed regret matching

---

**Input**: EFG game $G$, node $h$, policy $\sigma$, cumulative policy $\overline{\sigma}$, reach probabilities $T_i$, cumulative reach probability $Q$, cumulative regret $R$

**Output**: utility $u$

  **function** Traversal(Trav as a short form)

    $u = 0$

    **if** $V(h) = 0$ **then**

      **for** legal action $a$ in $A(h)$ **do**

        $T' = T; T_0' = T_0 * P(a|h)$

        $u+ = P(a|h) * Trav(G, \{h, a\}, \sigma, \overline{\sigma}, T', Q, R)$

      **end for**

      **return** $u$

    **end if**

    get the information set $I$ of node $h$

    $\delta = 1/(a + b * Q(I))$

    **for** legal action $a$ in $A(h)$ **do**

      action probability $p = \delta/|A(h)| + (1 - \delta)\sigma(a|I)$

      $T_i' = T_i; T_{V(h)}' = T_{V(h)} * p$

      action utility $u_a = Trav(G, \{h, a\}, \sigma, \overline{\sigma}, T', Q, R)$

      $u = u + p * u_a$

    **end for**

    $p_c = \left(\prod_{n \in N} T_n\right) / T_{V(h)}$

    $Q(I) = Q(I) + p_c * T_{V(h)}$

    **for** legal action $a$ in $A(h)$ **do**

      $R(a|I) = R(a|I) + p_c * (u_a - u)$

      Update $R(a|I)$ if use CFR+ or DCFR

    **end for**

    $\sigma(I) = $ regret match$(R(I))$

    update average strategy $\overline{\sigma}(I)$

    **return** $u$

  **end function**

---

## 4.2 SEQUENTIAL PERTURB DECREASING

The CFR algorithm does not use the belief $\mu$, so we need to convert the formulation of sequential equilibrium. We notice that the belief $\mu$ is only used to compute the utility of the information set. If

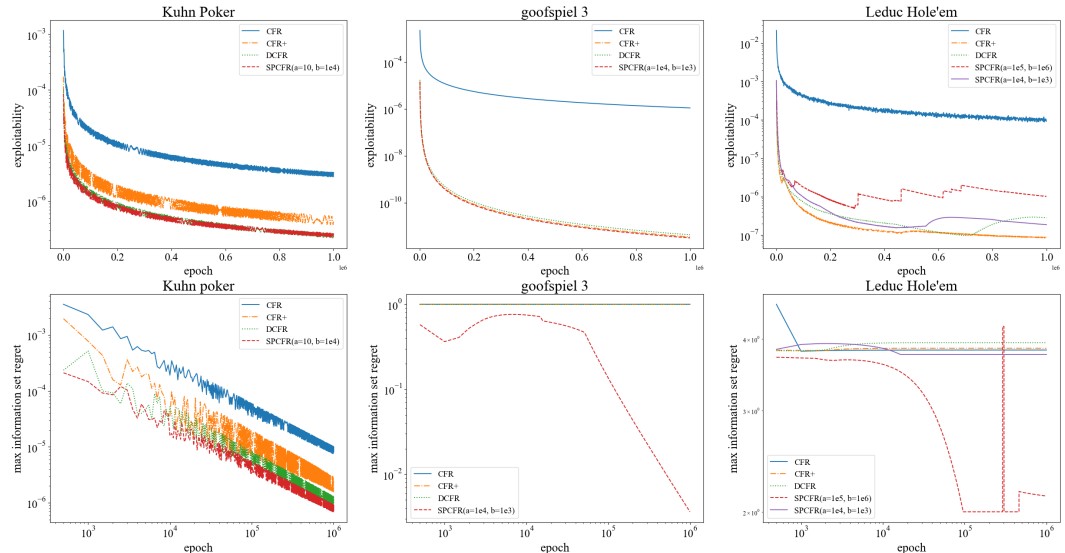

Figure 1: Comparison of CFR variants and SPCFR algorithm in three environments: Kuhn Poker, GoofSpiel, and Leduc Hold'em. The upper figures show the variation of exploitability with the number of nodes in the game tree visited by algorithms, and the lower figures show the maximum regrets of information sets.

---

**Algorithm 2** Sequential Restricted Counterfactual Regret Minimization
***
   set policy $\sigma$ uniform random policy,
   for every information set $I$, $Q(I) = 0, R(I) = 0, \overline{\sigma} = 0$
   **for** many episodes **do**
      for every player $i$, $T_i = 1$
      $Traversal(G, \varnothing, \sigma, \overline{\sigma}, T, Q, R)$
   **end for**
***

we use the pair $(\beta, u(\beta|\cdot))$, and there exists a series $(\beta^n, u^n(\beta|\cdot))$ that tends to $(\beta, u(\beta|\cdot))$, we can also show that this is a sequential equilibrium. Therefore, we only need to show that the algorithm can estimate $u(\beta|\cdot)$ accurately.

For those reachable information sets, if both players use the historical averaging strategy,

$$u\left(\overline{\beta}|I\right) = \mu_{\overline{\beta}}(h|I)u\left(\overline{\beta}|h\right) = \frac{P_{\overline{\beta}}(h)u\left(\overline{\beta}|h\right)}{P_{\overline{\beta}}(I)} = \frac{\frac{1}{T}\sum_{t=1}^{T} v(\beta^t|h)}{P_{\overline{\beta}}(I)} = C\sum_{t=1}^{T} v(\beta^t|h).$$

The counterfactual value is a constant multiple of the information set utility. For unreachable information sets, $P_{\overline{\beta}}(I) = 0$, the proof of the above would not be valid. We need some extra conditions to ensure the algorithm can converge to the sequential equilibrium.

**Theorem 2.** *The limit of average strategy $\overline{\sigma}$ is a sequential equilibrium, only if $\delta_I$ satisfied:*

*for all node $h$, the cumulative reach probability tends to infinity, $\lim_{T \to \infty} \sum_{t=1}^{T} P_{\beta_{-V(h)}^t}(h) = +\infty$.*

*And if $\lim_{t \to \infty} \beta^t$ exists, the condition above is sufficient and necessary.*

A full proof of Theorem 2 is provided in Appendix B. From this theorem, we can tell that any global method of perturbation reduction is unreasonable. Assume we use a global perturbation $\delta_T$, the reaching probability of each node at depth $h$ should not be less than $\delta_T^h$. However, according to Theorem 2, the cumulative reach probability should tend to infinity, which means $\delta_T$ must be $o(T^{-1/H})$. This implies that the convergence rate slows down rapidly as the depth of the game tree increases.

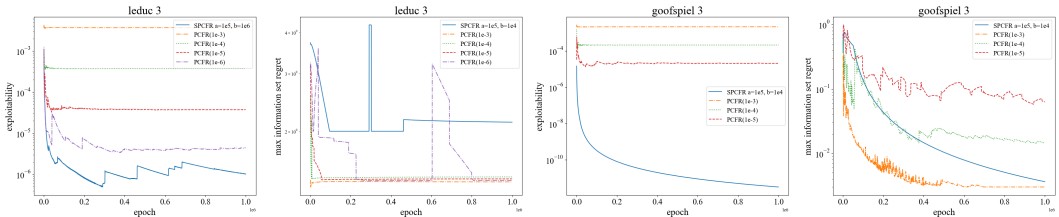

Figure 2: The performance of SPCFR and Discount CFR with various perturbs on Leduc Hold'em and GoofSpiel.

We give a local form definition of $\delta_T(I)$ to satisfy the follow conditions,

$$\delta_T(I) = \frac{1}{a + bQ^T(I)}, \tag{12}$$

where $Q^T(I) = \sum_{t=1}^{T} P_{\sigma^t}(I)$ is the cumulative reach probability. In particular, the cumulative reach probability of the root node is equal to the number of iterations. If $\delta_T(I)$ use this definition, we prove the upper limit of average regret of strategy $\sigma$ is $\overline{R_{\sigma,T}} \sim O(T^{-0.5})$.

Further, we will show that average strategy $\overline{\beta_T}$ and average strategy $\overline{\sigma_T}$ have the same limit when iteration tends to infinity. We can use substitute average strategy $\overline{\beta_T}$ for average strategy $\overline{\sigma_T}$. The average strategy $\overline{\beta_T}$ will converge faster because its action probability has not been restricted.

$$\lim_{T \to \infty} \frac{1}{T} \sum_{t=1}^{T} \sigma_i^t(I, a) = \lim_{T \to \infty} \frac{1}{T} \left( \sum_{t=1}^{T} \beta_i^t(I, a) - \sum_{t=1}^{T} \delta_t \beta_i^t(I, a) \right). \tag{13}$$

It can be easily seen that the limit of the second half is 0. Thus, we proved that the limit of average strategy $\overline{\beta_T}$ and average strategy $\overline{\sigma_T}$ is the same when the iteration tends to infinity.

$$\lim_{T \to \infty} \frac{1}{T} \sum_{t=1}^{T} \sigma_i^t(I, a) = \lim_{T \to \infty} \frac{1}{T} \sum_{t=1}^{T} \beta_i^t(I, a). \tag{14}$$

We split the algorithm into two parts for convenience. Algorithm 1 shows the game tree traversal part of the algorithm, and Algorithm 2 is the main loop.

## 5 EXPERIMENT

### 5.1 EXPERIMENT SETTING

We compared the effects of CFR Zinkevich et al. (2007), CFR+ Bowling et al. (2015), and DCFR Brown & Sandholm (2019) with our algorithm SPCFR in four environments: Kuhn Poker, Leduc Hold'em, and GoofSpiel. Here is a brief description of the four experiment environments.

**Kuhn Poker** is a simple imperfect information card game with only three cards: King, Queen, and Jack. In Kuhn poker, two players hold one private card and successively choose to bet or pass. If one player chooses to bet, the other player can only choose to bet or fold. If both players bet or pass, the player with the bigger card wins.

**Leduc Hold'em** is also a card game. It is a simplified version of Texas Hold'em, having only two betting rounds and six cards: two suits of King, Queen, and Jack. In the first round, each player holds a private card; in the second round, one public card is revealed.

**GoofSpiel** is a simple rules card game, but has a large number of information sets. In the full game, the player has 13 cards of the same suit, while another suit is used as the prize cards. Each round, a prize card is revealed, and two players simultaneously bid for it by selecting a card from their hand. The bidder with the highest number wins the prize card, if the numbers are the same, no one will

get the prize. All cards can only be used once. The game continues until all prize cards have been claimed, and the player who gets the highest prize wins.

We use Kuhn Poker, Leduc Hold'em, Liar's Dice, and Goofspiel(3 cards) environments in Open-SpielLanctot et al. (2019) with default parameters. All experiments were run on a Linux server with four Intel Xeon Platinum 8268 processors. Each experiment in the three environments took about 30 min, 3 h, and 13 h, respectively.

## 5.2 Experiment Result

To compare the performance of algorithms, we employ two evaluation metrics: exploitability and maximum information set regret. **Exploitability** is a widely used evaluation metric that describes the distance between the current strategy and the NE. Maximum information set regret reflects the worst-case deviation from the optimal action at any information set in the game tree, describing how closely the strategy approximates the EFPE. We have not found a metric that describes how close a strategy is to SE. Since EFPE is included in SE, we use this metric to evaluate the effectiveness of our algorithm.

Figure 1 compares the performance of SPCFR and CFR variants in Kuhn Poker, GoofSpiel, and Leduc Hold'em. Except for the Kuhn Poker game, we can see that all other CFR variants have stabilized at a fixed value for the maximum information set regret in the other two environments. It is to be expected that the CFR algorithm and its variants do not converge to EFPE. Kuhn Poker is a simple game with only 12 information sets. There are no unreachable information sets in the NE, so other algorithms are also able to reduce the maximum information set regret value. Nevertheless, our algorithm, SPCFR, still has the fastest decrease among them. For the exploitability metric, compared with the best variant of the CFR algorithm, SPCFR works approximately or slightly better. This suggests that a reasonable descent rate of perturbation does not limit the reduction of the exploitability.

In Figure 2 we compare our algorithm with perturbed CFR. The algorithm partially differs from the paper in that we add the perturbation to the DCFR rather than CFR+. Due to the perturbations, there is a lower bound on the exploitability in perturbed CFR, which is not present in our SPCFR algorithm. In the GoofSpiel environment, SPCFR outperforms both perturbations of $1e-4$ and $1e-5$ in the maximum information set regret metric, and is close to the perturbation of $1e-3$ in the end. The performance in Leduc hold 'em is slightly worse for SPCFR than for perturbed CFR in the maximum information set regret, but perturbed CFR also suffers from large fluctuations in PCFR when the perturbation is small($1e-6$).

## 6 Conclusion

This paper proposes the sequential perturbed counterfactual regret minimization(SPCFR) algorithm for computing Nash equilibrium refinement, sequential equilibrium, of imperfect-information games. We compare it with other NE computing algorithms, CFR variants, and the approximate EFPE computing algorithm, perturbed CFR. SPCFR follows the methods of perturbed CFR in the regret matching part, restricting the minimum value of the probability of each action for the strategy. Further, we extend the global perturbation to a local perturbation and give a suitable perturbation descent method. We give the necessary conditions for localized perturbation descent to converge to SE, and its proof is sufficient under certain assumptions. The experiment has shown that SPCFR converges close to the best CFR variant in terms of exploitability, and can also approach SE well compared to PCFR. In future work, we hope to improve other algorithms for solving imperfect-information games, such as PSRO, so that they can solve the sequential equilibrium.

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

# A PROVE OF THEOREM 1

We will prove Theorem 1 using the same method as the upper bound estimate of Gordon (2006) for cumulative regret, where we define $f(\mathbf{x}) = ([x_i]_+)$, $F(\mathbf{x}) = \sum_i [x_i]_+^2 / 2$. $f(\mathbf{x})$ should be the gradient of $F(\mathbf{x})$.

$$\mathbf{R_{t+1}} - \mathbf{R_t} = \mathbf{v_p^{\sigma^t}}(I) - \left( \left( \delta_t / N \cdot \mathbf{1} + (1 - \delta_t) \sigma_{\mathbf{p}}^{\mathbf{t}}(I) \right) \cdot \mathbf{v_p^{\sigma^t}}(I) \right) \cdot \mathbf{1}$$

$$(\mathbf{R_{t+1}} - \mathbf{R_t}) \cdot f(\mathbf{R_t}) = \delta_t \left( \left( \sigma_{\mathbf{p}}^{\mathbf{t}}(I) - \mathbf{1}/N \right) \cdot v_p^{\sigma^t}(I) \right) (f(\mathbf{R_t}) \cdot \mathbf{1})$$

$$\leq 2|u|_{max} \delta_t \left( f(\mathbf{R_t}) \cdot \mathbf{1} \right)$$

where $|u|_{max}$ is the upper bound of utility, $|v|_{max}$ is the upper bound of Counterfactual utility, $P_{\sigma^t}(I)$ is the reach probability of the information set $I$ when all players follow the strategy $\sigma^t$. The $F(\mathbf{x})$ satisfied,

$$F(\mathbf{x} + \Delta) \leq F(\mathbf{x}) + \Delta \cdot f(x) + \|\Delta\|_2,$$

$$F(\mathbf{R_{t+1}}) = F(\mathbf{R_t} + \mathbf{x_t})$$

$$\leq F(\mathbf{R_t}) + \mathbf{x_t} \cdot f(\mathbf{R_t}) + \|\mathbf{x_t}\|^2$$

$$\leq F(\mathbf{R_t}) + 2|u|_{max} \delta_t (f(\mathbf{R_t}) \cdot \mathbf{1}) + |u|_{max}^2 |A|.$$

$$\leq 2|u|_{max} \sum_{k=1}^{t} \delta_k (f(\mathbf{R_k}) \cdot \mathbf{1}) + |u|_{max}^2 |A| t$$

We know that $F(\mathbf{R_{t+1}}) \geq 2(\mathbf{R_t} \cdot \mathbf{p})^2$ for any legal probability $\mathbf{p}$ We can get the upper bound,

$$2(\max_a \mathbf{R_t})^2 \leq 2|u|_{max} \sum_{k=1}^{t} \delta_k (f(\mathbf{R_k}) \cdot \mathbf{1}) + |u|_{max}^2 |A| t$$

$$\max_a \mathbf{R_t} \leq \sqrt{|u|_{max} \sum_{k=1}^{t} \delta_k (f(\mathbf{R_k}) \cdot \mathbf{1}) + |u|_{max}^2 |A| t / 2}$$

$$O(R_t(I, a)) = O \left( \sqrt{\sum_{k=1}^{t} O(\delta_t) O(R_k(I, a)) + t} \right).$$

Assume $O(\delta_t) = O(t^\alpha), O(R_t(I, a)) = O(t^\beta)$,

$$O(t^\beta) = O \left( \sqrt{\sum_{k=1}^{t} O(t^{\alpha+\beta}) + t} \right)$$

$$= O(\sqrt{t^{\alpha+\beta+1} + t}).$$

It can be solved that $\beta = \max\{0.5, \alpha + 1\}$. The upper bound of information set regret is $\pi$ is $R_t(I, a) \sim O(t^{\max\{0.5, \alpha+1\}})$.

If $\delta_T(I) = \dfrac{1}{a + b\sqrt{Q_t(I)}} \sim \Theta\left(Q_t(I)^{-0.5}\right)$, and we assume the reach probability $P_{\sigma^t}(I) \sim \Theta(t^\gamma)$.

Of course, we need $\gamma \leq 0$. $Q_t(I) \sim \Theta(t^{\gamma+1})$ which means $\alpha = -\frac{\gamma+1}{2} < -0.5$. So we proved $\forall I$, $R_t(I, a) \sim O(t^{0.5})$, which means the upper bound of the total average regret is $\overline{R_t} \sim O(t^{-0.5})$.

# B PROVE OF THEOREM 2

The proof of CFR has already shown that if the average regret $\overline{R_\sigma^T}$ tends to 0, in the reachable information sets, the average strategy $\overline{\sigma^T}$ tends to the sequential equilibrium. For those unreachable information sets, we need to satisfy the definition of sequential rationality and consistency of the sequential equilibrium.

According to the definition of *consistent*, we have equation,

$$u(\sigma|I) = \sum_{h \in I} \mu(h|I)u(\sigma|h) = \sum_{h \in I} \lim_{\sigma^n \to \sigma} \frac{P_{\sigma^n}(h)}{P_{\sigma^n}(I)} u(\sigma|h).$$

In the CFR method, we use counterfactual values $v(\sigma|I) = \sum_{h \in I} \lim_{T \to \infty} \frac{1}{T} \sum_{t=1}^{T} v(\sigma^t|h)$ to estimate utility. Since the regret and utility values differ by a constant factor, we need to ensure that for any two nodes in the same information set, the ratio of their utilities is equal to the ratio of counterfactual values. And once the consistency is satisfied, the regret matching will satisfy the sequential rationality. The equation we need to prove is,

$$\lim_{\sigma^n \to \sigma} \frac{P_{\sigma^n}(h_1)u(\sigma|h_1)}{P_{\sigma^n}(h_2)u(\sigma|h_2)} = \lim_{T \to \infty} \frac{\sum_{t=1}^{T} P_{\sigma^t_{-V(h_1)}}(h_1)u(\sigma^t|h_1)}{\sum_{t=1}^{T} P_{\sigma^t_{-V(h_2)}}(h_2)u(\sigma^t|h_2)}.$$

In fact, in the information set of player $i$, the history action of player $i$ in all nodes on the same information set must be the same. Otherwise, it can distinguish between these nodes. We can leave the player's actions out of the equation. Expand the average utility $u$,

$$\lim_{\sigma^n \to \sigma} \frac{P_{\sigma^n}(h_1)u(\sigma|h_1)}{P_{\sigma^n}(h_2)u(\sigma|h_2)} = \lim_{\sigma^n \to \sigma} \lim_{T \to \infty} \frac{\sum_{t=1}^{T} P_{\sigma^n_{-V(h_1)}}(h_1)u(\sigma^t|h_1)}{\sum_{t=1}^{T} P_{\sigma^n_{-V(h_2)}}(h_2)u(\sigma^t|h_2)}.$$

Here we need to assume that,

$$\lim_{t \to \infty} u(\sigma^t|h_1) < \infty,$$
$$\lim_{t \to \infty} u(\sigma^t|h_2) < \infty,$$

Less than infinity indicates that the limit exists. We believe this assumption is universal. Then we can use the O'Stolz theorem,

$$\lim_{\sigma^n \to \sigma} \frac{P_{\sigma^n}(h_1)u(\sigma|h_1)}{P_{\sigma^n}(h_2)u(\sigma|h_2)} = \lim_{\sigma^n \to \sigma} \lim_{t \to \infty} \frac{P_{\sigma^n_{-V(h_1)}}(h_1)u(\sigma^t|h_1)}{P_{\sigma^n_{-V(h_2)}}(h_2)u(\sigma^t|h_2)},$$

$$\lim_{T \to \infty} \frac{\sum_{t=1}^{T} P_{\sigma^t_{-V(h_1)}}(h_1)u(\sigma^t|h_1)}{\sum_{t=1}^{T} P_{\sigma^t_{-V(h_2)}}(h_2)u(\sigma^t|h_2)} = \lim_{t \to \infty} \frac{P_{\sigma^t_{-V(h_1)}}(h_1)u(\sigma^t|h_1)}{P_{\sigma^t_{-V(h_2)}}(h_2)u(\sigma^t|h_2)}.$$

Let $\lim_{t \to \infty} \frac{P_{\sigma^t_{-V(h_1)}}(h_1)u(\sigma^t|h_1)}{P_{\sigma^t_{-V(h_2)}}(h_2)u(\sigma^t|h_2)} = A$, $\lim_{t \to \infty} \frac{P_{\sigma^n_{-V(h_1)}}(h_1)u(\sigma^t|h_1)}{P_{\sigma^n_{-V(h_2)}}(h_2)u(\sigma^t|h_2)} = B(\sigma^n)$.

$$\forall \delta_1 > 0, \exists M_1 > 0, s.t. \forall t > M_1, \left| \frac{P_{\sigma^t_{-V(h_1)}}(h_1)u(\sigma^t|h_1)}{P_{\sigma^t_{-V(h_2)}}(h_2)u(\sigma^t|h_2)} - A \right| < \delta_1.$$

$$\forall \delta_2 > 0, \exists M_2 > 0, s.t. \forall t > M_2, \left| \frac{P_{\sigma^n_{-V(h_1)}}(h_1)u(\sigma^t|h_1)}{P_{\sigma^n_{-V(h_2)}}(h_2)u(\sigma^t|h_2)} - B(\sigma^n) \right| < \delta_2.$$

Let $n > \max\{M_1, M_2\}$, we have $|A - B(\sigma^n)| < \delta_1 + \delta_2$. It means that $\lim_{n \to \infty} B(\sigma^n) = A$.

