# OpenReview forum: "Counterfactual Regret Minimization for Sequential Equilibrium"
_ICLR.cc/2026/Conference — Submitted to ICLR 2026_

### Official Review · Reviewer_ZX7G · 2025-10-21

**Soundness:** 2
**Presentation:** 2
**Contribution:** 2
**Rating:** 4
**Confidence:** 4

**Summary:**

This paper introduces the Sequential Perturbed Counterfactual Regret Minimization (SPCFR) algorithm, an iterative algorithm designed to compute refinements of the Nash Equilibrium (NE), specifically the Sequential Equilibrium (SE), in two-player zero-sum imperfect-information games. The experiments show that SPCFR converges close to the best CFR variant in terms of exploitability, and can also approach SE well compared to Perturbed CFR.

**Strengths:**

- Important Problem: The paper tackles the important and challenging problem of computing equilibrium refinements beyond the standard NE. While NE strategies can prescribe suboptimal actions in parts of the game tree that are unreachable in equilibrium, refinements like SE provide a more robust solution concept.
- Novel Approach: Existing algorithms typically employ global local perturbation, while this paper use perturbation. The paper provides a convincing argument that a global perturbation that decays over time would lead to impractically slow convergence as the game tree depth increases, as the reach probability of deep nodes would diminish too quickly. This provides a strong rationale for the proposed local perturbation.

**Weaknesses:**

- The primary weakness of this paper is its lack of clarity. Upon my initial review, I assumed that this paper aims to addressed the problem of learning an EFPE. However, this paper only demonstrates the convergence to an SE.

- Symbol definitions are unclear. For instance,  $a$, $b$, and $\sigma$ in Equation (12) lack the definitions.

- The symbol definitions differ significantly from those in most CFR papers and the closely related work--Farina et al. [2017], which increases the difficulty for readers.

**Questions:**

- In the experiments, how were the hyperparameters $a$ and $b$ for the perturbation schedule in Equation 12 selected for each game?

- Why you do not compare with PCFR and PCFR+ (Predictive CFR and Predictive CFR+)? I notice that you use “PCFR” to denote Perturbed PCFR. However, “PCFR” usually denotes Predictive PCFR.

- Why not conduct comparisons on larger games, such as HUNL subgames?

- Why is there no established convergence theory indicating that SPCFR+ converges to SE? Is it due to that some assumptions cannot be satisfied?

---

> ### Author Response · Authors · 2025-11-27
>
> Dear Reviewer ZX7G,
>
> Thank you for your thorough and insightful review. We sincerely appreciate the time and effort you dedicated to evaluating our work. Below, we provide a point-by-point response to each of the comments and concerns you raised.
>
> **Q4-1: Clarity of Paper**
>
> **A4-1:** Thank the reviewer for the helpful comments. In the revised manuscript, we will reduce the discussion of EFPE and explicitly state that the objective of this paper is to compute SE, in order to minimize potential misunderstandings.
>
> **Q4-2: Unclear symbol definitions**
>
> **A4-2:** We were really sorry for our careless mistakes. In the final version, we will carefully check all the notations to ensure that each symbol has a clear definition. In this paper, our notation follows the conventions of the original CFR algorithm (Zinkevich et al., 2007). Thanks again for your careful reading.
>
> **Q4-3: About the selection of hyperparameters $a$ and $b$.**
>
> **A4-3:** We agree that clearer guidance on hyperparameter selection is important. In our experiments, we set the values of $a$ and $b$ empirically. Concretely, $a$ is chosen roughly as the inverse of the early-iteration exploitability, while $b$ is selected within the range $0.01a$ to $100a$. In the revised manuscript, we will include sensitivity plots in the appendix.
>
> **Q4-4: Comparison with Predictive CFR**
>
> **A4-4:** We appreciate the reviewer’s suggestion.
> In fact, SPCFR is a method designed for computing Nash equilibrium refinements, which accounts for the possibility that players may make mistakes. The regret-matching improvements introduced by SPCFR are orthogonal to many CFR variants, including Predictive CFR. In the revised manuscript, we will compare SPCFR with Predictive CFR.
>
> Exploitability in Leduc 3
> |algorithm|1e3|1e4|1e5|1e6|
> |---|---|---|---|---|
> |CFR|1.182e-2|1.829e-3|4.431e-4|9.313e-5|
> |CFR+|9.373e-4|**5.416e-6**|**4.726e-7**|8.831e-8|
> |DCFR|**2.404e-4**|6.973e-6|7.219e-7|2.833e-7|
> |Predictive CFR|5.934e-4|1.352e-5|1.055e-6|**7.231e-8**|
> |SPCFR|2.917e-4|2.217e-5|1.050e-5|6.04e-6|
>
> Max Information Set Regret in Leduc 3
> |algorithm|1e3|1e4|1e5|1e6|
> |---|---|---|---|---|
> |CFR|**3.808**|3.825|3.826|3.827|
> |CFR+|3.820|3.853|3.854|3.854|
> |DCFR|3.813|3.936|3.942|3.942|
> |Predictive CFR|3.921|3.958|3.958|3.958|
> |SPCFR|3.888|**3.811**|**1.091**|**0.094**|
>
> **Q4-5: Comparison on large games (such as HUNL subgames).**
>
> **A4-5:** We agree that our primary benchmarks are relatively small.
> To address scalability concerns, we will include additional experiments on Leduc-5 and Goofspiel-4. However, it is difficult to compute the maximum information-set regret in HUNL, because it requires a full traversal of the game tree.
>
> Exploitability in Leduc 5
> |algorithm|1e3|1e4|1e5|2e5|
> |---|---|---|---|---|
> |CFR|9.408e-3|1.562e-3|2.823e-4|1.500e-4|
> |CFR+|2.768e-4|7.180e-6|2.980e-7|1.446e-7|
> |DCFR|3.231e-4|8.818e-6|5.302e-7|3.021e-7|
> |SPCFR|2.306e-4|2.913e-5|1.490e-5|1.247e-5|
>
> Max Information Set Regret in Leduc 5
> |algorithm|1e3|1e4|1e5|2e5|
> |---|---|---|---|---|
> |CFR|13.09|11.00|11.00|11.00|
> |CFR+|**4.128**|4.523|4.414|4.409|
> |DCFR|4.298|**4.443**|4.397|4.393|
> |SPCFR|8.504|5.651|**1.094**|**0.770**|
>
> **Q4-6: About the convergence theory**
>
> **A4-6:** The difficulty in establishing SE convergence for SPCFR does not stem from unsatisfiable assumptions. Standard CFR provides convergence guarantees only for **average-iterate** strategies, whereas SE computation methods typically rely on **last-iterate** guarantees. Consequently, no existing theoretical framework directly implies that SPCFR converges to an SE. Demonstrating such convergence would require proving that the sequence of **average strategies** generated by SPCFR satisfies the SE definition. The core technical challenge is therefore to sufficiently control the **last-iterate** behavior so that the induced **average strategy** sequence attains the limits required for SE.

---

### Official Review · Reviewer_dG2m · 2025-10-28

**Soundness:** 2
**Presentation:** 2
**Contribution:** 2
**Rating:** 6
**Confidence:** 4

**Summary:**

This paper introduces the Sequential Perturbed Counterfactual Regret Minimization (SPCFR) algorithm, which extends the CFR framework to converge toward sequential equilibrium (SE) in imperfect-information sequential games. The key innovation is the use of local, decreasing perturbations at each information set, rather than a fixed global perturbation. The authors provide theoretical guarantees on regret bounds and convergence to SE under certain assumptions, and demonstrate empirically that SPCFR outperforms existing CFR variants and perturbed CFR in terms of exploitability and maximum information set regret in games like Kuhn Poker, Leduc Poker, and GoofSpiel.

**Strengths:**

-	The idea of using local, decreasing perturbations is novel and well-motivated, addressing limitations of fixed global perturbation methods.

-	Theoretical analysis is provided, including regret bounds and convergence proofs under specific assumptions.

-	Empirical evaluation uses standard metrics and games, showing consistent improvements over relevant baselines.

**Weaknesses:**

-	The theoretical convergence guarantee relies on the strong assumption that the limit of the strategy sequence exists, which is not sufficiently justified or discussed.

-	Empirical evaluation is limited to small-scale games, with no demonstration of scalability to larger or more complex domains.

-	Performance gains in some environments (e.g., Leduc Poker) are modest, raising questions about the practical significance of the improvements.

-	Lack of ablation studies or sensitivity analysis for key parameters (e.g., a and b) weakens the reproducibility and practical utility of the method.

-	The comparison with existing methods is incomplete, omitting recent non-CFR-based approaches for equilibrium refinement.

**Questions:**

-	Could the local perturbation scheme be integrated with sampling-based CFR variants (e.g., MCCFR) to improve scalability? If so, what would be the expected trade-offs?

-	Under what practical conditions does the assumption of a strategy sequence limit hold? Are there common game structures or settings where this assumption is likely to fail?

-	Have the authors considered evaluating SPCFR in larger games?

---

> ### Author Response · Authors · 2025-11-27
>
> Dear Reviewer dG2m,
>
> We feel great thanks for your professional review work on our article. As you are concemed, there are several problems that need to be addressed. According to your nice suggestions, we have made extensive corrections to our previous draft, the detailed corrections are listed below.
>
> **Q3-1: About the Assumption of Theorem 2.**
>
> **A3-1:** We sincerely appreciate your valuable suggestion. The limit-existence assumption is required only for sufficiency,
> while the necessity part of Theorem 2 does not depend on it.
> Empirically, we believe that SPCFR can still converge to SE even without this assumption. In the revised manuscript, we will explicitly separate the sufficiency and necessity arguments to avoid confusion.
>
> **Q3-2: About more experiments on large games.**
>
> **A3-2:** Thanks for your suggestion.
> More experiments on large games (Leduc 5 and Goofspiel 4) have been reported. Please refer to **Q1-3** to Reviewer ccH4.
>
> **Q3-3: Performance Gains is modest in Leduc games.**
>
> **A3-3:** There are two evaluation metrics, exploitability and max information set regret. CFR and its variants perform well in minimizing exploitability but struggle to reduce maximum information set regret. The primary aim of SPCFR is to minimize the max information set regret while maintaining exploitability at a level comparable to other CFR variants. According to the experiments, our SPCFR has significant improvement in maximum information set regret against existing CFR based methods.
>
> **Q3-4: Lacking sensitivity analysis for parameters $a$ and $b$.**
>
> **A3-4:** Thanks for your construtive suggestion.
> The hyperparameters $a$ and $b$ respectively control the scale and decay exponent in $\delta(I)=\frac{1}{a+bQ(I)}$. Empirically, smaller values of $a$ and $b$ accelerate the reduction of the maximum information-set regret but slows down the decrease in exploitability. Larger $a$ and $b$ have the opposite effect.
> We will discuss the sensitivity in the final version.
>
> Exploitability in Leduc 3
> |$a$|$b$|1e3|1e4|1e5|1e6|
> |---|---|---|---|---|---|
> |1e2|1e4|3.631e-4|4.486e-5|4.178e-5|4.143e-5|
> |1e4|1e4|6.606e-4|2.217e-5|1.050e-5|6.042e-6|
> |1e6|1e4|**2.853e-4**|**9.085e-6**|**8.313e-7**|**4.881e-7**|
> |1e4|1e2|8.263e-4|1.084e-4|9.688e-5|7.731e-5|
> |1e4|1e6|9.162e-4|1.098e-5|1.332e-6|7.384e-7|
>
> Max Information Set Regret in Leduc 3
> |$a$|$b$|1e3|1e4|1e5|1e6|
> |---|---|---|---|---|---|
> |1e2|1e4|**3.873**|3.431|1.115|0.133|
> |1e4|1e4|3.889|3.811|1.091|0.094|
> |1e6|1e4|3.896|3.896|6.206|6.922|
> |1e4|1e2|3.883|**1.533**|**0.017**|**0.001**|
> |1e4|1e6|3.895|4.370|3.775|1.102|
>
> **Q3-5: Other non-CFR-based approaches for equilibrium refinement.**
>
> **A3-5:** Thanks for your nice suggestion.
> We compare our SPCFR with a non-CFR-based method named OOMD (Bernasconi et al., 2024) in Leduc3. The results are as follow. We can see that, our SPCFR outform OOMD in both metrics.
>
> Exploitability in Leduc 3
> |algorithm|1e3|1e4|1e5|2e5|
> |---|---|---|---|---|
> |OOMD|2.351|2.330|2.246|2.199|
> |SPCFR|**2.917e-4**|**2.217e-5**|**1.050e-5**|**8.408e-6**|
>
> Max Information Set Regret in Leduc 3
> |algorithm|1e3|1e4|1e5|2e5|
> |---|---|---|---|---|
> |OOMD|10.67|10.51|10.14|9.969|
> |SPCFR|3.888|**3.811**|**1.091**|**1.376**|
>
>
> **Q3-6: About combining SPCFR with sampling-based CFR.**
>
> **A3-6:** SPCFR can naturally extend to sampling-based traversals. A practical modification is to treat the contribution of each sampled action to $Q(I)$ as 1, rather than weighting by the action probability $p$. This estimator remains unbiased, since the expected contribution of an action in each iteration is exactly $p$. The main trade-offs are increased estimator variance and greater jitter in local $Q(I)$ estimates, which can make $\delta(I)$ noisier.

---

### Official Review · Reviewer_Prr9 · 2025-10-29

**Soundness:** 2
**Presentation:** 1
**Contribution:** 2
**Rating:** 2
**Confidence:** 5

**Summary:**

This paper proposes Sequential Perturbed Counterfactual Regret Minimization (SPCFR), an improved CFR-based algorithm for computing sequential equilibria in imperfect-information games. By introducing local perturbations that decay adaptively during training, the method overcomes the limitations of fixed-perturbation approaches like Perturbed CFR. The authors provide theoretical guarantees that SPCFR converges to sequential equilibrium under certain conditions and demonstrate experimentally on Kuhn Poker, Leduc Hold’em, and GoofSpiel that it achieves comparable exploitability to the best CFR variants while significantly reducing information-set regret.

**Strengths:**

The main strength of this paper lies in its meaningful motivation and direction. Extending CFR toward computing sequential equilibria addresses an important yet relatively underexplored problem. The proposed idea of using locally adaptive perturbations to refine equilibrium computation is conceptually appealing and shows potential for further development.

**Weaknesses:**

1. The front matter (Introduction, Related Work, Preliminaries) feels disproportionately long for a 9-page submission—the paper does not present its core contribution until the end of page 5. It would help to surface the main ideas earlier (e.g., add a brief Motivation section) and consider moving parts of Related Work to the appendix.
2. The figures need substantial improvement.
  - In Figs. 1–2, text is too small, and the lines in Fig. 2 lack clarity.
  - Using a log scale for the top row in Fig. 1 could make trends clearer.
  - In Fig. 1, “Goofspiel” appears inconsistently (capitalization) and I am unsure what the “3” in the title denotes.
  - The sharp jumps in the Leduc plot (especially the lower-right panel of Fig. 1) likely indicate an implementation issue; this deserves investigation rather than being described as mere “fluctuation.”
  - In addition, the parameters “$a$” and “$b$” for SPCFR are not explained in the text (they only appear in Algorithm 1).
3. I do not fully understand the necessity of fixing the functional form $\delta_T(I)=\frac{1}{a+b\,Q(I)}$. Theoretically, $\delta_T(I)$ only needs to be sufficiently small; a simpler or smaller schedule might suffice. Also, if the pseudocode is literal, then $Q(I)$ may change across different nodes within the same infoset in single iteration, implying $\delta_T(I)$ is not consistent within that iteration—please clarify whether this is intended.
4. The paper mentions Liar’s Dice (line 435) but does not include it in the experimental evaluation.
5. The experimental scope is relatively limited (which may be due to hardware or the full-traversal design). It would help to state explicitly in the main text why only a small set of environments is used, or to scale up (e.g., larger-deck Kuhn/Leduc/Goofspiel). Including more recent baselines such as PCFR [1] would also strengthen the comparisons.
6. The presentation could be more intuitive. Since the target equilibrium accounts for dominated strategies, it would be informative to show a case study—for example, in Leduc—where a player deviates into an off-path/dominated strategy and compare SPCFR’s response to CFR/DCFR.

---

[1] Farina G, Kroer C, Sandholm T. Faster game solving via predictive blackwell approachability: Connecting regret matching and mirror descent[C]//Proceedings of the AAAI Conference on Artificial Intelligence. 2021, 35(6): 5363-5371.

**Questions:**

Refer to the previous section

---

> ### Author Response · Authors · 2025-11-27
>
> Dear Reviewer Prr9,
>
> Thank you for your thorough and insightful review. We will try our best to improve the manuscript and make some changes to the manuscript. Below, we provide a point-by-point response to your main concerns.
>
> **Q2-1: About the sharp jumps.**
>
> **A2-1:** Thanks for your professional analysis.
> We investigated the reported sharp jumps and found they were caused by numerical instability when computing cumulative regret. Specifically, we stored cumulative utility and per-action cumulative utility separately and computed regret as their difference. When these quantities become large, subtraction introduces floating-point precision errors that can be amplified after division by small reach probabilities. We have corrected this mistake and re-run the experiments. The results are reported as follow.
>
> Max Information Set Regret of SPCFR in Leduc 3.
> | |**2.91e6**|**2.92e6**|**3.02e6**|**3.03e6**|
> |---|---|---|---|---|
> |before fixing|2.000|4.210|4.208|2.000|
> |after fixing|1.174|1.170|1.088|1.083|
>
> **Q2-2: Make clear the hyperparameters of $a$ and $b$.**
>
> **A2-2:** Thanks for your suggestion. The hyperparameter $a$ sets the initial perturbation scale and prevents the policy from remaining near uniform in early iterations.
> The hyperparameter $b$ controls the decay exponent of the local perturbation $\delta(I)=\frac{1}{a + b*Q(I)}$.
> The choice of $a$ and $b$ has no effect on the convergence of our method, but it does influence on the convergence speed.
>
> **Q2-3: About Fixing the Functional Form of Perturbation.**
>
> **A2-3:** Thank you for pointing this out.
> The local perturbation $\delta(I)=\frac{1}{a+bQ(I)}$ is to achieve a balance between exploitability and max information set regret. In reachable information set, the perturbation $\delta\sim O(t^{-1})$ does not effect the exploitability. In unreachable information set, the perturbation decreases slowly to minimize the information set regret.
> If we choose a sufficiently small perturbation, the max information set regret will decrease sufficiently slow[1].
> If using a global perturbation $\delta_t$ that depends only on the iteration $t$, we should ensure the cumulative reach probability of the deepest information sets tends to infinity. It implies that $\delta_t$ cannot decay faster than $t^{-1/D}$, where $D$ is the max depth of the game tree. Such a decay significantly slows convergence.
>
> [1]Gabriele Farina et al., Regret Minimization in Behaviorally-Constrained Zero-Sum Games, in ICML 2017
>
> **Q2-4: About the Updating of Q(I) in Pseudocode**
>
> **A2-4:** We were really sorry for our careless mistakes.
> $Q(I)$ should be updated only once per iteration instead of being updated inside the traversal function. We will fix the pseudocode in the revision. Thank you for your reminder again.
>
> **Q2-5: About more experiments on scalability.**
>
> **A2-5:** Thanks for your suggestion. The additional experimental results on large games (Leduc 5 and Goofspiel 4) are reported in **Q1-3** to Reviewer ccH4. Also, we compare SPCFR with Predictive CFR baseline, as shown in the following table.
>
> Max Information Set Regret in Leduc 3
> |algorithm|1e3|1e4|1e5|1e6|
> |---|---|---|---|---|
> |$CFR$|**3.808**|3.825|3.826|3.827|
> |$CFR+$|3.820|3.853|3.854|3.854|
> |$DCFR$|3.813|3.936|3.942|3.942|
> |$Predictive \ CFR$|3.921|3.958|3.958|3.958|
> |$SPCFR$**(ours)**|3.888|**3.811**|**1.091**|**0.094**|
>
>
> **Q2-6: About the case studies of off-path/dominated strategy.**
>
> **A2-6:** Thanks for your suggestion. To study how strategy behavior change in off-path/dominated information sets between SPCFR and other CFR varients, we compare the performance of different algorithms on two off-path information sets in Leduc 3. The results are as follow.
>
> |history|algorithm|fold|call|raise|
> |---|---|---|---|---|
> |*JccJr|$CFR$|7.135e-7|0.273|0.727|
> ||$CFR+$|2.378e-12|0.348|0.652|
> ||$DCFR$|1.143e-12|0.302|0.698|
> ||$PCFR$|3.333e-5|3.333e-5|1.000|
> ||$SPCFR$|2.756e-5|2.756e-5|1.000|
> |K*rcQrr|$CFR$|0.043|0.957|0|
> ||$CFR+$|0.495|0.505|0|
> ||$DCFR$|0.349|0.651|0|
> ||$PCFR$|1.000|5.107e-5|0|
> ||$SPCFR$|1.000|3.796e-5|0|
>
> History refers to the complete sequence of events, including the cards dealt to each player, the public cards revealed, and all player actions. JQK denote card ranks, * represents the opponent’s unknown hole card, and c/r indicate call and raise, respectively.
>
> In the information set “*JccJr”, we hold a pair of Jacks, a guaranteed winning hand. After the opponent makes an irrational raise, raising rather than calling yields a higher payoff. Because once they call, we win a larger pot. Thus, raising offers a marginal advantage, and both PCFR and SPCFR correctly prefer it.
>
> In the “K*rcQrr” information set, we irrationally raise twice with only hole card K and board card Q. Then, the opponent continues to raise, the correct response is to fold immediately to minimize losses, as their aggression strongly suggests they hold a Q in their hole card.

---

### Official Review · Reviewer_ccH4 · 2025-11-03

**Soundness:** 2
**Presentation:** 2
**Contribution:** 2
**Rating:** 4
**Confidence:** 4

**Summary:**

This paper introduces the Sequential Perturbed Counterfactual Regret Minimization (SPCFR) algorithm, an iterative method designed to compute the sequential equilibrium (SE) in two-player zero-sum games. The core problem is that NE, which standard algorithms like CFR compute, can prescribe suboptimal actions in parts of the game tree that are unreachable under the equilibrium policy. SPCFR addresses this by building on the idea of a perturbed game, where all actions have a non-zero probability of being played. The key contributions are a method that uses a local perturbation at each information set, and a specific decreasing schedule for this perturbation that is dependent on the cumulative reach probability of that set. The authors provide a theoretical analysis suggesting the algorithm converges to an SE and present experimental results on several small games (Kuhn Poker, Leduc Hold'em, GoofSpiel) showing its performance relative to CFR variants and a fixed-perturbation baseline.

**Strengths:**

1. The paper tackles an important and well-motivated problem. The shortcomings of NE in extensive-form games are well-known, and developing scalable, iterative algorithms for computing refinements like SE is a valuable research direction.

2. The core idea of using a local perturbation that anneals based on the information set's visit coun is clever and intuitive. It elegantly adapts the exploration to parts of the game tree as they are encountered, which is a more principled approach than a global, fixed perturbation.

3. The inclusion of theoretical analysis (Theorems 1 and 2) to justify the algorithm's convergence properties is a significant strength. Providing proofs for convergence to SE, even under specific assumptions, lends substantial credibility to the proposed method.

**Weaknesses:**

1. The main claim of the paper is that SPCFR is an effective iterative method for computing SE. However, the experiments primarily compare against algorithms that compute NE (CFR, CFR+, DCFR). The most critical comparison would be against other iterative algorithms that aim for equilibrium refinements, such as the OOMD-based approach by Bernasconi et al. (2024) mentioned in the related work. Without this, it is difficult to assess the practical advantages of SPCFR over the state-of-the-art in this specific subfield. The comparison in Figure 2 is against a "perturbed CFR", but this seems to be a baseline implemented by the authors rather than a well-established algorithm from prior work.

2. The motivation for an iterative approach to SE is its potential scalability to large games where methods based on solving linear programs are infeasible. However, the experiments are conducted on very small benchmark games (Kuhn Poker, Leduc Hold'em, 3-card GoofSpiel). These games are not large enough to convincingly demonstrate the scalability and practical necessity of the proposed algorithm. A stronger case would be made by evaluating SPCFR on a game that is known to be challenging for non-iterative solvers.

3. Clarity on Theoretical Assumptions: Theorem 2 provides a necessary and sufficient condition for convergence that requires the cumulative reach probability of every node h to tend to infinity. This is a very strong condition. It is not immediately obvious how the proposed perturbation schedule guarantees this for nodes that are deep in the tree or are only reachable via multiple "mistakes". A more detailed discussion of how this assumption is met in practice would strengthen the paper.

**Questions:**

1. Could the authors elaborate on why other iterative methods for computing equilibrium refinements, such as the OOMD algorithm, were not included as experimental baselines? A direct comparison would be very informative.

2. The paper's motivation rests heavily on the need for scalability. Could you discuss the feasibility of applying SPCFR to a larger game, and why such an experiment was not included in the current version?

3. Regarding the condition in Theorem 2, how sensitive is the practical satisfaction of this condition to the choice of hyperparameters $a$ and $b$ in the perturbation schedule? Does a poor choice risk non-convergence for certain information sets?

---

> ### Author Response · Authors · 2025-11-26
>
> Dear Reviewer ccH4,
>
> Thanks for your thorough and insightful review. We sincerely appreciate the time and effort you dedicated to evaluating our work. Below, we provide a point-by-point response to each of the comments and concerns you raised.
>
> **Q1-1: About the comparison with other iterative methods.**
>
> **A1-1:** Thanks for your suggestion.
> Computing refinements of the Nash Equilibrium (NE) is important to exploit opponents' mistakes in the real world. Though CRF is a powerful tool for computing NE, there are only little attention paid for computing NE refinements based on the CFR framework. This paper aims to bridge this gap. So, it mainly compares SPCFR with CFR and its variants. We also compare our SPCFR with OOMD method (Bernasconi et al., 2024) in Leduc 3. The results are  reported as follow.
>
> Exploitability in Leduc 3.
> | |1e3|1e4|1e5|2e5|
> |---|---|---|---|---|
> |$CFR$|$1.182e-2$|$1.829e-3$|$4.431e-4$|$2.556e-4$|
> |$CFR+$|$9.373e-4$|$5.416e-6$|$4.726e-7$|$2.005e-7$|
> |$DCFR$|$2.404e-4$|$6.973e-6$|$7.219e-7$|$3.605e-7$|
> |$OOMD$|$2.351$|$2.330$|$2.246$|$2.199$|
> |$SPCFR$|$2.917e-4$|$2.217e-5$|$1.050e-5$|$8.408e-6$|
> | | | | | |
>
> Max Information Set Regret in Leduc 3.
> ||1e3|1 e4|1e5|2e5|
> |---|---|---|---|---|
> |$CFR$|$3.808$|$3.825$|$3.826$|$3.827$|
> |$CFR+$|$3.820$|$3.853$|$3.854$|$3.854$|
> |$DCFR$|$3.813$|$3.936$|$3.942$|$3.942$|
> |$OOMD$|$10.67$|$10.51$|$10.14$|$9.969$|
> |$SPCFR$|$3.888$|$3.811$|$1.091$|$1.376$|
> | | | | | |
>
>
> **Q1-2: Clarification on the “perturbed CFR” baseline in Fig.2**
>
> **A1-2:** Sorry for the confusion. The baseline named "perturbed CFR" follows the algorithmic description in [1].
> That paper does not provide a canonical algorithm name, so we used the descriptive label “perturbed CFR” for simplicity. We will add an explicit citation and a short footnote to avoid the confusion.
>
> [1] Gabriele Farina et al., Regret Minimization in Behaviorally-Constrained Zero-Sum Games, in ICML 2017.
>
> **Q1-3: Further experiments on large games.**
>
> **A1-3:** Thanks for this valuable suggestion to evaluate our method on large games. Our results on large games (Leduc 5 and Goofspiel 4) also suport our conclusion. For larger game such as HUNL, it is difficult to compute the evaluation metric - maximum information-set regret. Moreover, the potential scalability of SPCFR we claimed is that it can be combining with sampling-based CFR varients(such as MCCFR, DeepCFR).
>
> Evaluation on exploitability in Leduc 5
> |algorithm|1e3|1e4|1e5|2e5|
> |---|---|---|---|---|
> |$CFR$|$9.408e-3$|$1.562e-3$|$2.823e-4$|$1.500e-4$|
> |$CFR+$|$2.768e-4$|$7.180e-6$|$2.980e-7$|$1.446e-7$|
> |$DCFR$|$3.231e-4$|$8.818e-6$|$5.302e-7$|$3.021e-7$|
> |$SPCFR$|$2.306e-4$|$2.913e-5$|$1.490e-5$|$1.247e-5$|
>
>
> Evaluation on Max Information Set Regret in Leduc 5
> |algorithm|1e3|1e4|1e5|2e5|
> |---|---|---|---|---|
> |$CFR$|$13.09$|$11.00$|$11.00$|$11.00$|
> |$CFR+$|$4.128$|$4.523$|$4.414$|$4.409$|
> |$DCFR$|$4.298$|$4.443$|$4.397$|$4.393$|
> |$SPCFR$|$8.504$|$5.651$|$1.094$|$0.770$|
>
>
> **Q1-4: Clarity on Theoretical Assumptions.**
>
> **A1-4:** Let the cumulative reach probability of parent information set $Q(I)$ tends to infinity.
> Under our local perturbation $\delta(I)=\frac{1}{a+bQ(I)} \geq CQ^{-1}(I)$, the cumulative reach probability of child nodes $Q'(I)\geq C\sum P(I)Q^{-1}(I)$.
> We can scale $Q(I)$ to the smallest $2^k\geq Q(I)$, then $Q'(I)\geq C\sum_k2^{-k}\sum_{2^{k-1}<Q(I)<2^k} P(I)$.
> Since $Q(I)$ is the sum of reach probabilities $P(I)$, we have $\sum_{2^{k-1}<Q(I)<2^k} P(I)\approx2^{k-1}$.
> Substituting this approximation gives $Q'(I)\geq C\sum_k 2^{-k}2^{k-1}=Ck/2\rightarrow\infty$.
> The root node of the game tree clearly satisfies the assumption $Q(I)$ tends to infinity.
> Therefore, all the nodes in the game tree satisfy the condition in Theorem 2.
>
>
> **Q1-5: About the choice of  $a$ and $b$.**
>
> **A1-5:** As we show in above **A1-4**, the condition is not sensitive to the choice of hyperparameters $a$ and $b$, and always holds for all $a\geq 1$ and $b>0$.
> However, the choice of $a$ and $b$ has a significant impact on the convergence speed of our $SPCFR$ method, as shown in the following tables.
>
> Exploitability in Leduc 3 with different choices
> |$a$|$b$|1e3|1e4|1e5|1e6|
> |---|---|---|---|---|---|
> |$1e2$|$1e4$|$3.631e-4$|$4.486e-5$|$4.178e-5$|$4.143e-5$|
> |$1e4$|$1e4$|$6.606e-4$|$2.217e-5$|$1.050e-5$|$6.042e-6$|
> |$1e6$|$1e4$|$2.853e-4$|$9.085e-6$|$8.313e-7$|$4.881e-7$|
> |$1e4$|$1e2$|$8.263e-4$|$1.084e-4$|$9.688e-5$|$7.731e-5$|
> |$1e4$|$1e6$|$9.162e-4$|$1.098e-5$|$1.332e-6$|$7.384e-7$|
>
> Max Information Set Regret in Leduc 3 with different choices
> |$a$|$b$|1e3|1e4|1e5|1e6|
> |---|---|---|---|---|---|
> |$1e2$|$1e4$|$3.873$|$3.431$|$1.115$|$0.133$|
> |$1e4$|$1e4$|$3.889$|$3.811$|$1.091$|$0.094$|
> |$1e6$|$1e4$|$3.896$|$3.896$|$6.206$|$6.922$|
> |$1e4$|$1e2$|$3.883$|$1.533$|$0.017$|$0.001$|
> |$1e4$|$1e6$|$3.895$|$4.370$|$3.775$|$1.102$|

---

### Meta-Review · Area_Chair_mkHx · 2025-12-30

**Summary:**

The main concerns of the Reviewers are about clarity and the experimental evaluation of the proposed algorithm. As for the latter, the main critiques raised by the Reviewers are: (i) the experimental evaluation is carried on relatively small game instances, (ii) missing comparison with very related algorithms (such as the OOMD variant by Bernasconi et al. (2024))., and (iii) some "jumps" in the plots showing regret performance. The Authors have partially addressed such concerns in their rebuttals. However, some concerns till remain bout the experimental evaluation and the comparison with existing algorithms (see the following section of the meta-review).

Additionally, one other concern raised by the Reviewers is the fact that it is **not** sufficiently clear that the algorithm proposed in the paper converges somehow to a sequential equilibrium, and **not** to an extensive-form perfect equilibrium. This is somehow misleading since sequential equilibria are much weaker than extensive-form perfect equilibria. Moreover, I think that converge guarantees of the algorithm (Theorem 2) are quite "opaque", as they rely on an assumption that appears to be rather strong, and it is not easy to establish whether it can be effectively met in practical cases.

For the reasons above, I believe that the papers does **not** pass the bar for acceptance at ICLR.

**Reviewer Concerns:**

The Authors partially addressed the Reviewer's concerns about the experimental evaluation. In particular, they provide some experiments on larger games (only Leduc 5) and they provide a comparison with the OOMD variant by Bernasconi et al. (2024). Moreover, they claimed that hey fixed some numerical instability issues in the computation of the regret that were the cause of the "jumps" in the plots observed by the Reviewers. As for the comparison with the OOMD algorithm by Bernasconi et al. (2024), I am still not fully convinced by the preliminary experimental results presented by the Authors in their rebuttal, since the algorithm by Bernasconi et al. (2024) seems to perform extremely worse than the one proposed in the paper and the "vanilla" versions of CFR as well, which is perhaps more concerning. Indeed, this is in stark contrast with the experimental results presented (Bernasconi et al., 2024), where their OOMD algorithm seems to greatly outperform the "vanilla" versions of CFR. This makes me doubtful about the actual implementation of the algorithm by Bernasconi et al. (2024) that the Authors implemented. My doubts are further exacerbated by the fact that, in the "Related Works" section, the Authored say that the algorithm by Bernasconi et al. (2024) "...needs to be optimized on the overall game matrix", which is **not** true. Indeed, after having a look at the paper by Bernasconi et al. (2024), I am quite certain that their OOMD algorithm can indeed be implemented efficiently by means of an iterative procedure (see Section 5 in the paper by Bernasconi et al. (2024)).

**Reviewer Scores:**

Reviewer ccH4, Score: 4 – I believe that the rebuttal would not have changed the reviewer’s opinion, especially given the other reviews.

Reviewer Prr9, Score: 2 – I believe that the rebuttal would not have changed the reviewer’s opinion, especially given the other reviews.

Reviewer dG2m, Score: 6 – I believe that the rebuttal would not have been sufficient to persuade this reviewer to champion the paper and convince the other reviewers to accept it.

Reviewer ZX7G, Score: 4 – I believe that the rebuttal would not have changed the reviewer’s opinion, especially given the other reviews.

---

### Decision · Program_Chairs · 2026-01-26

Reject